# Integrating Microfluidics and Electronics in Point-of-Care Diagnostics: Current and Future Challenges [note 1]

**DOI:** 10.3390/mi13111923

**Published:** 2022-11-07

**Authors:** Valerio Francesco Annese, Chunxiao Hu

**Affiliations:** 1Center for Nano Science and Technology@PoliMi, Istituto Italiano di Tecnologia, 20133 Milan, Italy; 2James Watt School of Engineering, University of Glasgow, Glasgow G12 8QQ, UK

**Keywords:** microfluidics, point-of-care, electronics, CMOS, review, personalized medicine

## Abstract

Point-of-Care (POC) diagnostics have gained increasing attention in recent years due to its numerous advantages over conventional diagnostic approaches. As proven during the recent COVID-19 pandemic, the rapidity and portability of POC testing improves the efficiency of healthcare services and reduces the burden on healthcare providers. There are hundreds of thousands of different applications for POC diagnostics, however, the ultimate requirement for the test is the same: sample-in and result-out. Many technologies have been implemented, such as microfluidics, semiconductors, and nanostructure, to achieve this end. The development of even more powerful POC systems was also enabled by merging multiple technologies into the same system. One successful example is the integration of microfluidics and electronics in POC diagnostics, which has simplified the sample handling process, reduced sample usage, and reduced the cost of the test. This review will analyze the current development of the POC diagnostic systems with the integration of microfluidics and electronics and discuss the future challenges and perspectives that researchers might have.

## 1. Introduction

Point-of-care (POC) technology refers to all miniaturized, portable, and automatized devices capable of providing healthcare close to or near the patient [1,2,3,4,5]. In practice, POC platforms are portable diagnostic devices that can be operated by the general population in any location, including at home, in an ambulance, in hospitals, critical care facilities and remote locations [6,7,8,9]. POC testing is a new, emerging healthcare model, and has been developed and expended dramatically over the last 30 years [3]. At present, the most commonly used approach for testing, in healthcare throughout the world, is the centralized laboratory [1]. Typically, samples are collected by trained personnel from various locations, including general practice surgeries and hospitals. Samples are then transferred to a laboratory where they are analyzed by trained personnel. Results are then communicated to the patient. The entire process is time consuming and requires specialists. The use of POC devices simplifies the process of sample testing by providing an on-the-spot, sample-to-answer test in a few minutes [10]. POC provides results rapidly, thereby saving time that would otherwise be spent with transferring samples to the laboratory. There is no need to wait for a trained personal to run the tests and, thus, the results do not need to be transmitted and collected. POC platforms can therefore reduce the response of a test from hours/days to minutes [11].

The rapidity and portability of POC testing might be more advantageous than laboratory testing in specific applications. The rapidity of POC testing can make a difference between life-or-death for applications requiring immediate availability of diagnostic data, such as sudden and acute medical conditions [11]. For instance, sepsis survival rate improves by 7.6% per hour of earlier diagnosis [12]. For acute cardiovascular events, such as ischemic stroke, early intervention within the so-called golden period (1–2 h after the event) improves the survival rate by 80% [13]. The portability of POC platforms has the potential to improve healthcare quality in rural and remote areas [1]. Testing infectious diseases in resource-poor locations, for example, has the potential to save many lives by providing clinical information for conditions that otherwise go undiagnosed [1]. The need for a rapid, adaptable and low-cost POC testing platform, providing reliable and quick results have been outlined in pandemic scenarios [14]. The recent COVID-19 pandemic required population-wide strategies, including mass-testing and contact tracing, both potentially deliverable using POC technologies, and challenging to implement when adopting centralized testing [14]. POC technology has the potential to reduce medical costs in some applications as it requires small sample volumes, small reagents volumes, and low running costs. POC devices can also be easily applied to wearable and smart systems [15,16].

POC diagnostic devices are complex systems as they integrate all testing steps ‘from sample to answer’ [17]. Typically, a POC system includes a user interface, a sample delivery device, a reagent storage strategy, a reaction cell, sensors to detect the measurement reaction, a control and communication system, and data management storage [1]. This paper focuses on the sample handling system, which is typically achieved by microfluidics and its integration with sensors and readouts (electronics). The present work illustrates the design, manufacturing, and characterization of the microfluidic system. The work also discusses the relevant state of the art on microfluidics manufacturing and integration with electronics. We also provide additional readings for other parts of POC systems not analyzed in this work. The paper is organized as follows: Section 2 illustrates the main theoretical equation for passive microfluidics channels, focusing primarily on the design parameters. Section 3 discusses the most commonly used methods for microfluidics fabrication and provides a comparison between those presented. Section 4 describes the main challenges that limit the wide-spread integration of microfluidics and electronics. Section 5 compares recent successful integrations of microfluidics with electronics for POC systems. Section 6 illustrates the main applications of POC systems for diagnostics. Finally, Section 7 discusses the future steps required to make this technology pervasive in the market of commercial POC diagnostic systems.

## 2. Microfluidics Theory

Microfluidics is the study of microstructures capable of handling small quantities of fluids. Many microfluidic structures have been successfully used for a range of fluidic operations in lab-on-chip platforms [10,18,19,20,21,22]. Microfluidic channels are microstructures that confine the fluid and allow it to move in a controlled path. Microfluidic elements for controlling the flow of the fluid in the microchannel have been developed, including pumps (active and passive) and valves [23,24,25,26]. Microfluidic elements are usually combined to create microfluidic networks. Currently, microfluidic networks can reach very high complexity level integrating channels, valves, pumps and mixers [27]. One of the most frequent processes and applications in the microfluidic sector is mixing. In a microscale microfluidic chip, micro-mixing refers specifically to uniformly combining two or more types of fluids [23,28,29,30,31,32]. Commercial devices typically use capillary microchannels with a rectangular cross-section and with no external pumps or mixers. Therefore, this review focuses on microfluidic theory for the capillary and laminar flow regime. Additional resources for an overview of microfluidic elements are suggested here [23,33,34]. Detailed reviews on micro-mixing theory and applications in [32,35,36] are also recommended.

In microfluidic structures, the flow is primarily laminar, meaning that the behaviour of the liquid can be decomposed into a series of infinitesimal layers, flowing on top of each other without mixing. The Reynolds number (Re) is typically used to define the flow regime in a microfluidic structure. The Reynolds number is defined as the following:(1)Re=Inertial ForcesViscous Forces =ρulµ=ulν
where ρ is the fluid density (kg/m^3^), u is the velocity of the fluid in the structure (m/s), l is a characteristic linear dimension of the structure (m/s), µ is the dynamic viscosity of the fluid (Pa⋅s), and ν is the kinematic viscosity of the liquid (m^2^/s) [37]. For a microstructure, l ≈ 10^−6^ so Re < 1. Turbulent flow is present when Re > 4000, while when Re < 2000 the flow is laminar.

The study of the fluid kinematics is usually carried out using the Navier-stokes equation. A generic particle with mass m and velocity v is influenced by several independent forces (F_j_):(2)$$m\frac{\mathrm{dv}}{\mathrm{dt}}=\underset{j}{\sum }F_{j}\;\to V^{-1}m\frac{\mathrm{dv}}{\mathrm{dt}}=V^{-1}\underset{j}{\sum }F_{j}\;\to {\;\rho D}_{t}v=\underset{j}{\sum }f_{j}\;\;\underset{j}{\sum }f_{j}=\{\begin{array}{c} \rho \delta_{t}v_{x}\;\;\;\;\;\;\;\;\;\;\;\;\;\;\;\;\;\;\;\;\;\;\;\;\;\;\;\;\;\;\;\;\;\;\;1D\;Flow\\ \rho \{\delta_{t}v+\left(v\;\cdot \;\nabla \right)v\}\;\;\;\;\;\;\;\;\;\;\;\;\;3D\;Flow \end{array}$$
where V is the considered volume, f is the force density and D_t_ is the material time-derivative defined as [34]:(3)$$D_{t}=\{\begin{array}{c} \;\;\;\;\;\;\;\;\;\;\;\;\;\;\;\;\;\;\;\;\;\delta_{t},\;\;|\;1D\;Flow\;\\ \delta_{t}+\left(v\;\cdot \;\nabla \right),\;\;|\;3D\;Flow \end{array}$$

The final form of the Navier-Stokes equation can be calculated by inserting the complete expression for the force densities:(4)$${\rho \delta }_{t}v_{x}=-\delta_{x}p+\;µ\left(\delta_{y}{}^{2}+{\;\delta }_{z}{}^{2}\right)v_{x}+{\;f}_{x}\;\;\;\;\;\;\;1D\;\mathrm{Flow}$$
(5)$$\rho \{\delta_{t}v+\left(v\;\cdot \;\nabla \right)v\}=-\nabla p+\nabla^{2}v+\left\{\rho g+{\;\rho }_{\mathrm{el}}E\right\}\;\;\;\;3D\;\mathrm{Flow}\;$$
where, in the second member, the first term is the pressure-gradient force density, the second term is the viscous force density, and the third term is the body force density.

One of the methods for resolving the Naiver-Stokes equation is represented by the Hagen-Poiseuille equation, valid in static conditions and in a rigid straight structure when a pressure gradient Δp (Pa) is applied [34]:(6)$$∆p=R_{h}\;Q$$
where R_h_ is the hydraulic resistance (kg/m^4^ s), and Q is the flow rate (mole of fluid passing through a section in a unit of time, m^3^/s). There is a formal equivalence between the Hagen-Poiseuille and the 2nd Ohm’s law. The hydraulic resistance depends on the geometry of the structure and on the viscosity of the fluid. Specifically, for rectangular channels with height h, length L and depth w the R_h_ is:(7)$$R_{h}=\frac{12\;\text{µ}L}{\left\{1-0.63\left(\frac{h}{w}\right)\right\}h^{3}w}$$

R_h_ is generally high for microfluidic structures due to height h and the width w having micrometric dimensions [34].

In the absence of externally applied pressure and with the channel height and width in the order of hundreds of micrometres, the liquid can spontaneously move due to cohesive forces within the liquid and adhesive forces between the liquid and its surroundings. This effect is commonly referred to as capillary action [38]. With reference to Figure 1, the capillary pressure gradient (Δp) is related to the property of the fluid and the geometry of the microchannel [34]:
(8)$$∆p=\;\gamma \left(\frac{\cos{ϴ}_{b}+\cos{ϴ}_{t}}{h}+\frac{2\cos{ϴ}_{s}}{w}\right)$$
where ϴ denotes the contact angle of the different materials employed and γ the surface tension. According to Equation (8), when w << h, the capillary pressure gradient depends only on w and the microchannel can even be left open [39,40]. Under the assumption of laminar, steady-state flow, and in the absence of gravitational effects, the position of the advancing liquid l(t) can be obtained by manipulating Equations (6–8) [34,41]:(9)$$l\left(t\right)=\;h\;\sqrt{\frac{\Delta p}{6\;\eta L}\left(1-0.63\frac{h}{w}\right)t}$$
where Δp is the capillary pressure gradient, R_h_ is the hydraulic resistance, Q is the flow rate, η is the dynamic viscosity, L is the microchannel length. Equation (9) can be used as a designing equation when developing capillaries.

## 3. Microfluidics Fabrication Techniques

The most commonly used techniques for microfluidics fabrication can be grouped into four categories: lamination, micromachining, direct writing, and moulding [27,42]. Microfluidics devices, fabricated using lamination, are created by a stack of independently cut layers, bonded together to form a microfluidic network [42]. Each layer is cut individually, and the employed technique usually defines the resolution of the device. Laser-cut laminated microfluidic devices typically offer a better resolution [42]. Lamination is a versatile technique; thus, a wide variety of materials have been used for the fabrication of laminated microfluidic devices, including paper [43], glass slides, polymers (e.g., Polymethyl methacrylate or PMMA, polycarbonate), and tapes [42]. In this category, the depth of the microfluidic channel can be tuned by controlling the thicknesses of the layers. The layers composing the laminated structure are typically bonded by thermal or adhesive bonding. Laminated devices offer several advantages, including rapid and straightforward process steps, as well as not needing cleanroom facilities, low-cost, versatility and scalability [42]. The main disadvantages of this technique are the difficulty in aligning the individual layers and the lower resolution when compared to alternative methods [42]. Despite its simplicity, lamination can support the fabrication of complex microfluidic chips, such as the one developed in [44]. In [44], for instance, the authors report the rapid prototyping of a haematology cartridge designed to determine red cell and platelet counts, haemoglobin concentration, a white cell differential count, and various derived parameters.

Microfluidics devices have also been fabricated using micromachining techniques [27,45]. The processes used for the fabrication of micromachined microfluidic devices are similar to the ones exploited for the fabrication of microelectromechanical systems (MEMS) and nanoelectromechanical systems (NEMS) [46]. Surface micromachining groups all the techniques, allowing for the fabrication of 2D and 3D microscale and nanoscale structures by deposition and removal of structural layers on a substrate. Examples of the processes commonly employed for the fabrication of surface micro-machined microfluidics are photolithography [45], e-beam lithography [45], and laser ablation [47].

Several materials can be used as substrates, including silicon [45], glass slides [45], and polymers [47]. When lithographic techniques are employed, the use of photoresist is widespread [45]. SU-8 is the most commonly used photoresist for surface micromachined microfluidic devices [42]. SU-8 is a common negative photoresist, performing high resolution, durability and a capacity for high aspect ratio structures [42]. Micromachined microfluidics devices can also be fabricated using etching techniques. This category of device is usually referred to as bulk micromachined [46,48]. This fabrication usually involves etching steps aimed to remove material from a bulk substrate, such as a silicon wafer [46]. In general, micromachined microfluidics has a higher resolution compared to laminated devices [42,45,46]. Micromachining can also be used for the fabrication of open channel devices [39,40]. Micromachined microfluidic devices also have drawbacks. Firstly, they require cleanroom facilities [42]. Materials and methods employed for the fabrication of these devices are sometimes costly and time-consuming. This also means the micromachined fabrication process is generally not repeatable on a large scale due to financial and timing considerations.

Another class of microfluidic devices can be fabricated using direct writing techniques, such as 3D printing [42], stereolithography [42], and two-photon polymerization [45]. Recently, 3D printed microfluidic devices have become increasingly popular, thanks to their low-cost and rapid fabrication. Printing also does not usually require cleanroom facilities. At present, 3D printing for the fabrication of fluidically sealed devices is available on the market [49].

Microfluidics devices can also be fabricated using a mould. The mould can be fabricated in many ways [42]. The resolution of the device usually depends on the technique adopted for the mould fabrication [42]. Arguably, photolithography is the most commonly used method for mould fabrication [42]. Although moulding encompasses several variations, moulded devices can be further divided into three categories: replica moulding, injection moulding and hot embossing [42]. They all have an initial stage of mould manufacturing [42]. Microfluidic devices fabricated by replica moulding employ a liquid polymer to be poured into the mould and subsequentially cured. The cured polymer is then peeled from the mould and bonded onto a glass slide or a substrate [42]. This fabrication process is also generally referred to as soft lithography [42]. Among the polymers employed for the fabrication of replica moulded devices, Polydimethylsiloxane (PDMS) is the most popular [42]. PDMS is a polymer structure with the repeating monomer units of SiO(CH_3_)_2_. It exhibits some advantages with respect to other materials used for microfluidic (e.g., PMMA, PCL) [48]. PDMS is transparent from 240 nm to 1100 nm, elastic, permeable to oxygen, and easy to use and to manipulate. Additionally, when freshly plasma-oxidized, it can be sealed to itself and other materials without any adhesive layer. In fact, under the exposure to oxygen plasma, the methyl groups (Si-CH_3_) on PDMS surfaces are attacked by reactive oxygen radicals and substituted by unstable silanol groups (Si-OH), which can permanently attach to ionic group on different plasma-oxidized substrates [50]. This property enables PDMS bonding directly on the target substrate without any intermediate adhesive layer. Finally, PDMS functionalization techniques are also robust and well-known [48,51,52,53,54,55,56,57,58].

In contrast, injection moulded microfluidics devices are fabricated by injecting a thermoplastic, melted into a liquid form, into the mould [42]. For injection moulding, usually two halves of the mould are used to create a cavity. Once the thermoplastic is cooled, the cast is removed from the mould. Similarly, in microfluidic devices fabricated using hot embossing moulding, a thermoplastic film is shaped onto the mould by applying pressure and heat [42]. Today, moulding plays a key role for custom developed fluidic platforms thanks to its simplicity, biocompatibility, and versatility [42]. Moulded microfluidic devices share similar challenges and limitations as those illustrated for micromachined systems. However, thanks to the reusability of the mould, this process is usually more scalable, and more promising for large scale production. Many additional material and methods have been operated for the fabrication of microfluidics devices. An extensive review of the materials and methods used for microfluidics is covered in these articles [42,59]. A comparison between widely adopted fabrication techniques for microfluidic systems is shown in Figure 2 and in Table 1.

## 4. Integration Challenge

Integrated platforms are significantly complicated to implement [60]. The integration of microfluidics and ICs is a gap so challenging to be bridged that some authors have questioned if this is possible [48]. The scientific literature shows that the most suitable method for microfluidics integration is application specific [17]. In some implementations, the combination of microfluidics and electronics can be not achieved at all [17].

There are three key challenges to be addressed when integrating CMOS chip with microfluidics: size compatibility, process compatibility, and economic considerations [61].

**Size compatibility.** The price of a CMOS chip is proportional to its area, so designers usually try to minimise the area [60,62]. Although fluidic channels have a compatible size with CMOS elements, fluidic input/output (I/O) ports need to be large enough (in the order of hundreds of micrometres) to allow practical operation. Increasing the area of CMOS to accommodate fluidic I/O in the design phase is possible. However, this typically requires an additional area, which yields an increased cost of the chip. The increase in cost is not acceptable with the respect to the affordability requirement. Furthermore, when the photoresist is applied by spin-coating on a millimetric area, surface tension creates an unwanted thicker ‘edge bead’ around the perimeter of the IC [60]. On millimetre-scale ICs, the bead can occupy the majority of the area and can pose a significant problem [60].

Size compatibility can be addressed by planarization [61]. Planarization allows for the integration of the CMOS chip into a larger substrate. Typically, fluidic I/O are incorporated onto the larger substrate, rather than onto the CMOS chip [61]. This technique has the potential to avoid increasing the area of the CMOS chip for microfluidic constraints which, in turn, would increase the cost of the CMOS chip. Notably, authors in [63,64,65,66] employ planarization before integrating the microfluidic network on top of the CMOS platform.

**Process compatibility.** This includes the necessity of a set of processes which demand new practical solutions [61]. Chip packaging is probably the most prominent complication to be overcome [61]. CMOS chips are usually connected to a chip package to be operated, and flip-chip bonding and wire-bonding are likely the two most reliable techniques for metallic interconnections [61]. Interconnects also require insulation and encapsulation [61]. In contrast to traditional electronic packaging, fluidic packaging has not been standardised by industry [60]. Thus, the approach to accommodate fluidics on CMOS is to either to modify a pre-existing standard package or to develop a custom package [60]. If wire-bonding is used for packaging, the fluidic network must avoid the bond-pads [60]. Consequently, the area the microfluidics network can occupy is largely decreased, and the geometry also constrained. Passivation of the wire-bonds can also be challenging in these conditions [60]. Remarkably, authors overcome the problem of wire bonds and metal interconnects by using liquid metal interconnects [67]. Thus, in [67], microfluidics ensures both sample handling and electrical connections. However, the approach has practical limitations and is not easily repeatable. Alternative techniques have also been adopted in the literature, such as screen-printing and additive manufacturing [61].

Material selection also poses a challenge to be addressed. Employed materials must be inert during the biological reaction and must not interfere [60]. The development of a reliable sterilization and cleaning method is also essential [60]. The use of materials such as PDMS and SU-8, which deteriorates over 200 °C, reduces the maximum temperature to which the platform can be exposed [60]. Furthermore, the wettability of materials needs to also be considered for the optimal flow and reduction of the evaporation [60]. Further complications concerning process compatibility also come from the topology of the IC, the alignment, and functionalisation [60,61].

**Economic considerations.** Microfluidics integration requires additional fabrication steps. However, POC microfluidic systems can be justified only when the production cost of the integrated system is low [61,68,69]. Consequently, this excludes several solutions which are not economically viable.

## 5. Microfluidics Integration with Electronics in POC

Overall, laminated devices are very difficult to monolithically integrate with IC, and they are more likely to be interfaced with electronics by other means [42]. Paper-based fluidics, among all the laminated devices, have been extensively used for several types of POC devices. Paper-based fluidics have many advantages when compared to other microfluidics platforms, such as its low-cost, easy functionalization process, and inherent flow control [46]. In a world where cell phones have outnumbered the global population, paper microfluidics have been used in conjunction with CMOS-based smartphone cameras [70]. However, they are usually aligned with the CMOS platform, but not monolithically integrated [46]. Monolithic integration is generally required for application aiming to detect a weak signal [57]. The integration eliminates any superfluous signal paths, which can additionally deteriorate the signal quality and introduce additional noise [57]. Additionally, monolithic integration reduces parasitic capacities and minimizes the footprint associated with sensing [57]. Micromachined microfluidic devices have a higher integration capability compared to laminated methods [46]. Specifically, it is possible to use an IC chip as a substrate and to monolithically integrate the fluidics on top of the device [46]. There are several ways of integrating microfluidics with electronics (Figure 3). 

In [73], for instance, the authors demonstrate a CMOS that is compatible microfluidic technology by integrating a microfluidic network on top of the optical biosensor devices. In this work, the microfluidics are integrated by spinning SU-8 on top of the sensing platform and performing a photolithography process [73]. A polymer slab is finally bonded onto the SU-8 microstructure to enclose the microchannel [73]. In [47], instead of SU-8, a photosensitive polymer PA-S321 from JSR company was used to create a microfluidic layer, as well as an adhesive layer. It was spin-coated and patterned onto the CMOS wafers using an EVG 6200 mask aligner. An ITO-coated glass substrate was used as a ceiling for the microfluidic structure. Ultrasonic machining was used to drill holes in the ITO-coated glass substrates to create inlet and outlet ports. The separately prepared CMOS part and glass part were then thermally bonded using an automated die/flip chip bonder. In [71], the author used epoxy to form open microfluidic channels on top of a CMOS sensor chip with the help of PDMA mould. In addition, the channels were finally enclosed with a thin layer of PVA-coated PDMS. Similar methods had been used in other works [63,65,66,74,75].

The integration of microfluidic networks fabricated with moulding techniques has also been reported in the literature. Authors in [67], for instance, adopt soft lithography to integrate a CMOS chip and microfluidic in a flexible package. In this research, two pieces of PDMS layer were separately moulded from patterned SU8 photoresist. The CMOS die was encapsulated by bonding those two pieces of PDMS layer. Finally, liquid metal was injected through the channels to form interconnects. In [76], to increase the adhesion of the moulded PDMS, a layer of polyimide was applied to the PDMS surface. Immediately upon completion of the plasma cleaning, the PDMS pieces were brought into contact with the surface modified polyimide PCB. After baking in an oven, a strong bond was formed. Similar approaches had been used by other researchers [48].

Laser engraving is another technique that has been widely used for fabricating microfluidic structures for the electronic devices. PMMA is one of the most common materials used for this method. For instance, authors in [66] created microfluidic channels on a piece of 2 mm thick PMMA sheet with laser engraving. The combination of the microfluidic channel and the FET chip, which embedded in epoxy, was achieved by using an interconnect layer of photoresist. Another material that is suitable for laser engraving is paper. In [43], open microfluidic channels were engraved using a CO_2_ laser onto a strip of paper, which was then encapsulated by a polymer sheet. Rather than bonding the microfluidic channels to the CMOS chip permanently, it was aligned on top of the chip by magnetic force. In this case, the CMOS chip can be reused.

In [64], replica moulding and laser engraving were combined to create a multi-layer PDMS microfluidic structure. The bottom PDMS layer, which was in contact with the IC chip, was fabricated by moulding from a SU-8 photoresist, and the top and intermediate PDMS layers were made by laser engraving. A 3D micromachined chip was finally built using a stamping and baking technique. Printing techniques can also be used to print structural materials, such us SU-8, on top of an IC device. For instance, authors in [72] demonstrated the integration of a CMOS device with microfluidics through direct writing. In this work, an organic ink is first deposited on top of the CMOS chip [72]. Subsequently, an optically clear epoxy resin is used to encapsulate the ink filaments and the CMOS device [72]. Finally, the ink filaments are extracted by applying heat and pressure, leaving epoxy-based microchannels on top of the CMOS chip [72]. Other techniques include wet etching on glass [77] and liquid phase dispersion on 2D fluid composites. Table 2 lists some of the research on microfluidic integration with electronics in POC.

## 6. Applications

Genomic, proteomic, and metabolic diagnostic devices have all been accomplished in a POC format using microfluidics and integrated circuits. Other applications not discussed in this review also include biophysical analysis [86], cell separation and sorting, material and drug delivery, drug testing, and organs-on-chip [55]. DNA and RNA are the targets of POC testing in genomics and transcriptomics [11]. Numerous methods have been proposed for detecting and amplifying the presence of nucleic acids [11]. The polymerase chain reaction, which creates billions of copies of a DNA sequence through iterative replications, is the most often employed method [87]. POC testing in genomics and transcriptome is especially useful for finding and identifying viruses, bacteria, fungi, microbes, pathogens, necrotic and abnormal cells [1], as well as other germs and pathogens. The integration of blood pre-treatment with DNA and RNA detection in a platform that is affordable, reliable, and user-friendly remains a significant challenge [1].

Proteomics POC testing focuses on proteins, such as enzymes, antibodies, and hormones [11]. The immunoassay technique, which uses antigen-antibody binding, is used in modern POC devices [11]. These assays focus on protein biomarkers for a multitude of diseases, including prostate cancer (PCa), cardiovascular disease (CVD), and bacterial and viral infections such as HIV, influenza, chlamydia, and hepatitis [11]. The enzyme-linked immunosorbent assay (ELISA) method is the foundation of the majority of protein analysis techniques [1]. The interaction of the target protein with the particular recognition molecule is shown in conventional ELISA testing using colorimetric or fluorescent readout signals [1]. ELISA analysis often involves a number of washing stages, which adds another layer of complexity to the design of a POC device [1]. ELISA can be implemented on both lateral flow assays (LFA) based POC and quantitative platforms [1]. It has been shown that an ELISA test based on LFA is simple to create [1]. However, multiplexed protein assays for qualitative platforms are now being developed [1,88]. For instance, utilizing a gold-nanoparticle assisted silver enhancement immunoassay, the authors of [88] suggest a CMOS-based device that can distinguish between blood samples having either, neither, or both rabbit anti-mouse (RAM) antibodies and/or anti-HIV antibodies. The suggested platform, according to the authors, is the first step toward developing a mass-manufacturable POC instrument that can quantify several proteins [88]. For PSA detection, numerous platforms have been created [89], including electrochemical [90], optical [91], cantilever-based [92], and other sensors [92], resulting in the first FDA-approved POC PSA test [93]. Additionally, POC systems for metabolic indicators have been created. The expense and size of the equipment generally employed for metabolite quantification are the key factors driving the development of POC platforms for metabolomics. The development of commercial colorimetric and fluorescence assay kits, for use in conjunction with a spectrophotometer, was also influenced by these reasons [94]. The current list of metabolites most frequently addressed are diverse and includes triglycerides, cholesterol, glucose, amino acids, choline, sarcosine and lactate. The scientific community is becoming increasingly interested in other metabolites, other than glucose meters, which are already well-established.

Both optical and electrochemical biosensors have been reported in the literature with a comparable performance for the quantification of L-type amino acids [95]. One of the most common targeted amino acids is glutamate, particularly given its association with neurological disorders [96]. Choline’s role in several disorders has sparked interest in quantifying it [97]. Similarly, the authors in [98], for example, use sarcosine oxidase to colorimetrically measure sarcosine in urine. The authors of this study show that the proposed assay can distinguish between PCa patients and the healthy population [98]. For lactate sensing, electrochemical and optical techniques have both been used [11,99]. Creatinine biosensors are also becoming increasingly popular due to the biomarker link with renal impairments [11]. POC platform monitoring lactate, cholesterol, triglycerides, and other lipids are also becoming increasingly popular for the management of CVDs [11]. The authors in [100] demonstrate the use of a CMOS sensor and an LED to provide comparable results to a commercial spectrophotometer for the colorimetric determination of bacterial concentrations. Similarly, the authors in [101] employ a CMOS sensor to quantify H_2_O_2_ using a colorimetric approach.

Multi-analyte platforms are also being developed [102]. The authors in [103], for instance, present a microfluidic lab-on-chip to quantify human body metabolites, using sub microliter droplets as reaction chambers. The authors demonstrate the suitability of the platform for glucose, glutamate, and pyruvate, individually [103]. The lab-on-chip takes advantage of an electrowetting chip, which transports and mixes the sample and the reaction for the initiation of a colorimetric reaction [103]. The reaction takes place in a microchannel fabricated by Teflon, perylene, and glass. PDMS microfluidic channels have been employed on the CMOS-based spectrophotometer system, as is reported in [104]. The system was used for the determination of glucose, uric acid, and cholesterol. A more complex PMMA-based system has been developed by the authors in [105] for the quantification of sorbic acid. Microfluidic channels supporting multiple tests have also been demonstrated for other applications including PCa [73] and sepsis [43].

The use of microfluidic systems has the potential to enable new hybrid technologies that can perform multiple tasks. Specifically, microfluidics has the potential to enable the manufacturing of independent reaction zones performing a test simultaneously, regardless of the nature of the test. Precision medicine could be revolutionized by multi-omics, which has the unique capacity to examine human samples at several levels and deliver combined diagnostic data. By merging multi-level omics approaches from the fields of genomes, proteomics, and metabolomics, multi-omics strategies in medical research hold promise for unravelling the complex biological processes [1]. The authors in [80] disclose an unprecedented device, in which the use of independent microfluidic reaction zones enable testing a metabolite and a protein, simultaneously, and on the same substrate for PCa. The authors in [106] also demonstrate that portable multi-omics systems can be implemented using electrochemical means.

## 7. Discussion and Future Directions

This paper provides an overview of the most recent efforts to integrate microfluidics and electronics into a portable system. Integrating microfluidics into POC devices is an ever-growing trend and many different approaches have been demonstrated thus far. According to the Web of Science [107], at present there are more than 1700 papers linked to the keywords “microfluidics” and “point-of-care”. Although the first work listed dates back to 1998 [108], more than one third of these works have been published since 2019 (see Figure 4a). The top five publishing institutions for the papers analysed on the Web of Science are: the University of California (4.915%), the Chinese academy of science (3.043%), Harvard University (2.984%), Duke University (2.984%), Stanford University (2.458%). However, 38.151% of the listed paper were published by American institutions. China and Germany follow with 16.384% and 7.197%, respectively (see Figure 4b). Resulting works are typically published from top publishers, including Elsevier (19.661%), Springer Nature (12.463%), IEEE (10.006%) and MDPI (9.421%). The top five funding agencies for the listed works are: the United States Department of Health Human Services (13.283%), the National Institutes of Health (NIH) USA (13.107%), the National Natural Science Foundation of China (NSFC) (10.532%), the National Science Foundation (NSF) (10.181%), and the European Commission (6.671%). The high volume of interest in the field is justified by the high societal and economic impacts of POC technology. As microfluidic POC devices became pervasive in our society, there are some challenges that need to be highlighted and addressed. One major concern about this technology remains the accuracy and LOD. Although there are POC platforms with LOD of micromolar fractions [71], efforts are needed to deliver reliable devices capable of high accuracy and extremely low LOD in a real-world scenario. Another main concern about this technology is in regard to materials. Most of the microfluidics devices developed thus far involve the use of PDMS. PDMS is a non-degradable material and is therefore not advised for use in single-use systems as it will accumulate in wastewater and in the environment [109,110].

Some works, such as [43], propose the use of paper-based microfluidics, which are a potential tool to improve the biodegradability of the platform. Other biodegradable materials, such as polycaprolactone (PCL) [111,112,113], are also potential candidate to be used as an alternative to PDMS.

As precision medicine enters the multi-omics era, technology also needs to evolve [80]. We envision the use of disposable biodegradable microfluidic devices, such as paper or other unexpensive degradable substrates, to address the concerns about waste accumulation in the environment in mass-scale production. We also foresee the use of low-cost, high-resolution sensors arrays [114,115,116] to improve the LOD of the system by improving the reliability of the platform and enabling the use of machine learning and artificial intelligence algorithms. In consideration of the technological solutions summarized in this paper, and the increasing interest in the field, we anticipate that the use of hybrid microfluidics–electronics systems in POC diagnostics will soon became pervasive.

## Figures and Tables

**Figure 1 micromachines-13-01923-f001:**
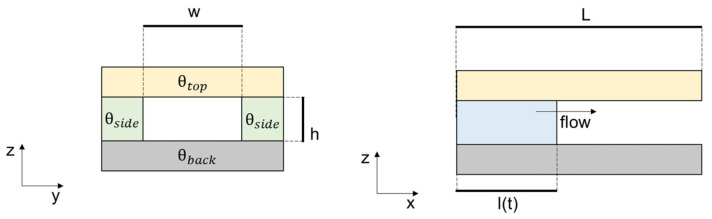
Schematic representation of a passive rectangular microfluidic channel.

**Figure 2 micromachines-13-01923-f002:**
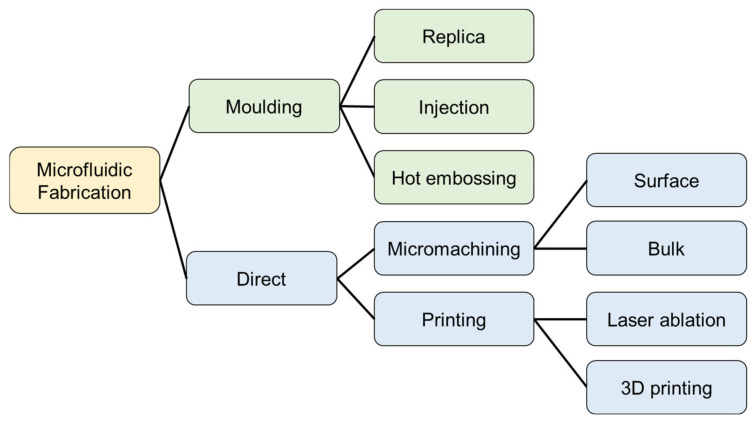
Main methods used for the manufacturing of microfluidic systems.

**Figure 3 micromachines-13-01923-f003:**
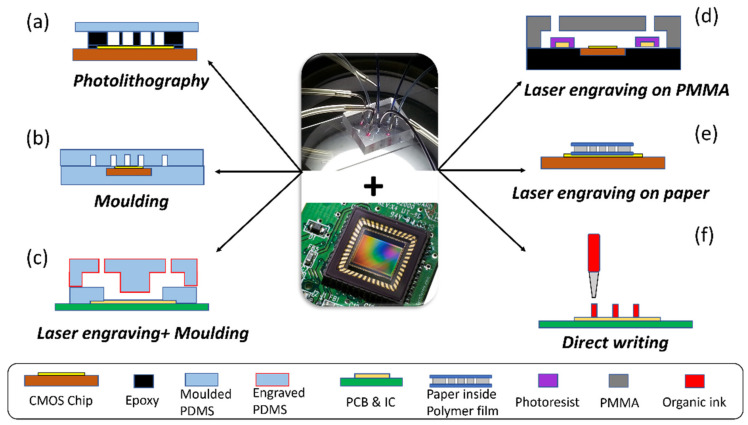
Examples of methods for integrating microfluidics with electronics. (**a**) Photolithography was used to fabricate microfluidic channels on top of a CMOS chip. Drawing was modified from [71]. (**b**) Moulding was used to fabricate microfluidic channels in PDMS, which encapsulated a CMOS chip. Drawing was modified from [67]. (**c**) Laser engraving was used to create PDMS microfluidic channels together with moulding. Drawing was modified from [64]. (**d**) Laser engraving was used to make microfluidic channels in PMMA, which was then attached to an epoxy substrate with embedded electronic chip. Drawing was modified from [66]. (**e**) Laser engraving was also used to generate microfluidic channels on paper strips, which was attached to a CMOS chip by magnetic force. Drawing was modified from [43]. (**f**) Direct writing was used to direct create patterns on a CMOS chip with organic ink, which could be washed away after encapsulation with epoxy. Drawing was modified from [72].

**Figure 4 micromachines-13-01923-f004:**
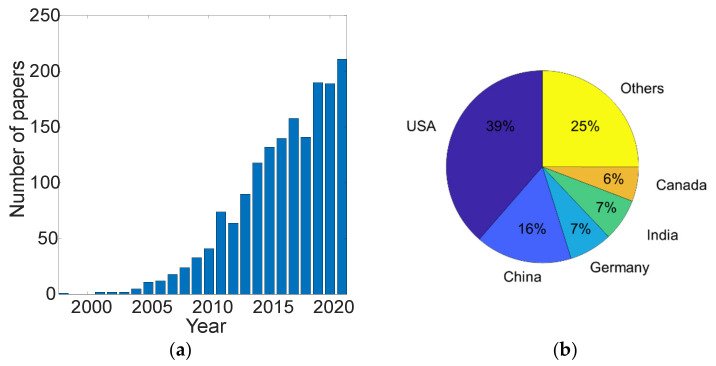
(**a**) Yearly published papers with both ‘microfluidics’ and ‘point-of-care’ tags from Web of Science. (**b**) Total published papers with both ‘microfluidics’ and ‘point-of-care’ tags from Web of Science according to geographical area.

**Table 1 micromachines-13-01923-t001:** Comparison between widely adopted fabrication techniques for microfluidic systems.

	Photolithography	Printing	Moulding
Resolution	µm	tens of µm	Down to µm scale (depending on the technique used for the fabrication of the mould)
Time to manufacture	- From hours to days - Several fabricating steps - Cleanroom facility needed	- From minutes to hours - Largely automatised - One device is fabricated at one time - No cleanroom facility needed	- Minutes * - Several devices are fabricated at one time - Can be automatised - No cleanroom facility needed *
Adaptability	- Wide range of substrates and structural materials - Channels have a rectangular cross-section	- Wide range of substrates and structural materials - 3D structures - Highly customisable	- Wide range of substrates and structural materials - Network topology depends on the technique used for the fabrication of the mould
Cost per device	High	Low	Low **
Suitable for large scale production	No (expensive and slow process)	No (slow process, lower resolution)	Yes

* After mould fabrication. ** When manufacturing a large number of devices.

**Table 2 micromachines-13-01923-t002:** Research on microfluidic integration with electronics in POC.

Technique	Target Substrate	Channels	Intermediate Layer	Distance to Substrate	Material	Ref.
Replica moulding	CMOS chip	4	No	0	Epoxy, PDMS	[71,74]
Replica moulding (moulding and encapsulation)	CMOS chip	n.d.	No	0	PDMS	[67]
Replica moulding (moulding and encapsulation)	CMOS chip and PCB	1	No	0	PDMS	[78]
Replica moulding (moulding and adhesive bonding)	ISFET and PCB	1	No	0	PDMS	[79]
Replica moulding (moulding and adhesive bonding)	CMOS and flexible PCB	4	Yes (Polyamide)	85 µm	PDMS	[76]
Replica moulding (moulding and plasma bonding)	IC chip	1	No	0	PDMS	[64]
Micromachining (encapsulation)	CMOS chip	2	No	0	Epoxy	[80]
Micromachining (encapsulation)	CMOS chip	1	No	0	Epoxy	[60,81,82,83]
Micromachining (photolithography and plasma bonding)	CMOS chip	n.d.	No	0	SU-8	[73]
Micromachining (planarization, photolithography)	CMOS chip	n.d.	Yes (ONO)	300 µm	SU-8, glass	[63]
Micromachining (planarization, photolithography and plasma bonding)	CMOS chip	1	No	0	SU-8, PDMS	[65]
Micromachining (photolithography and wet etching)	IC and flexible PCB	n.d.	Yes (PDMS)	120 µm	glass	[77]
Micromachining (photolithography and thermal bonding)	CMOS chip	1	No	0	JSR, glass	[84]
Micromachining (milling engrave and adhesive bonding)	CMOS chip and PCB	2	No	0	PMMA	[75]
Laser Engraving	CMOS chip	3	No	n.d.	Paper	[43]
Laser Engraving (planarization, laser engrave)	FET chip	n.d.	Yes (photoresist)	1.8 µm	PMMA	[66]
Direct writing	CMOS chip	1	No	0	Epoxy	[72]
Liquid Phase Dispersion	CMOS chip	1	No	0	2D-fluid composites	[85]

## Data Availability

Not applicable.

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
