# Peer review of "Integrating Microfluidics and Electronics in Point-of-Care Diagnostics: Current and Future Challenges†"

_micromachines, 2022, doi:10.3390/mi13111923_

Round 1

Reviewer 1 Report

This review first introduces the background and characteristics of point-of-care testing to illustrate its important applications in medical care. Then the application of microfluidics in point-of-care detection is introduced, and the contribution of micromixer in this respect is emphasized. Subsequently, the author conducts a theoretical analysis on the microfluidic flow to further illustrate the characteristics of microfluidic flow. Then, the author summarizes the manufacturing methods of microfluidic chip. The authors put forward the challenges of integrating microfluidic chips into an integrated platform, and then summarized the integration methods and characteristics of microfluidic chips and electronic devices in realizing even detection. This review is rich in content and comprehensive, but there are some minor problems need to be modified:

1. Microfluidic mixers are microstructures designed to favour the mixing of two different fluids.’Please check whether there is any error in this sentence.

2. Please check whether the unit of density in Formula 1 is written incorrectly

3. If necessary, please add some references in recent years.

Author Response

  1. We have now revised and rephrased this sentence. Please, see lines 87 – 90.

  1. We thank the reviewer for spotting the typo. We have now revised the text. Please, see line 101.

  1. We have now included 28 new references. Most of them have been published in the last 3 years. Please, refer to the reference section.

Reviewer 2 Report

1. For integrating of microfluidics and electronics in POC diagnostics, typical features of POC with sample pre-treatment, reaction, and detection functions are very critical. Therefore, the reviewer suggests that it is necessary to consider not only the microfluidics theory but also these parts for including in-depth review.

2. In microfluidics theory, as theories of mixing/fluidic control are also very important parts in microfluidics, the reviewer suggests that it is necessary to consider to make up these theories.

3. For potential readers, the reviewer recommends that clinical applications section might be included in this review to clarify importance of POC diagnostics technologies.

Author Response

  1. We agree with the reviewer that sample pre-treatment, reaction, and detection functions are very critical. However, we now clearly state in line 62 – 68 that this works focuses on the integration of microfluidics with electronics and we provide throughout the work additional reading for additional part of the system.

  1. We have now included suggested readings for mixing theory at microscale in Section 2. Please, see lines 93 – 94.

  1. In the revised version of the manuscript, we have now included a whole new section (Section 6) which focuses on POC applications. Please, see lines 386 – 465.

Reviewer 3 Report

The way authors outline the paper is quite approval, which islogical and comprehensive, and covers most aspects a reader would expect in the preliminary understanding of POC using microfluidics and electronics. A few comments to help improve the paper:

Table 1: the footnote number ‘1’ after ‘Minutes’ should be upscript. In the table, I suggest to change them to other marks rather than using numbers, to avoid the mixing with references through the paper.

Table2: To make the table more clear to look, I suggest to re-summarize and categorize the lists by either target substrate or technique.

In the last part, the authors mentioned about hybrid technologies, which is of great interests to readers. I would expect to see a brief summary of inter-discipline or hybrid study or applications in big picture, rather than one specific example.

In the conclusion part, right after challenges, it would be more improved if authors can use 1-2 sentences to propose the most valuable trends or practical solutions towards future POC microfluidics.

Author Response

  1. We have now revised Table I according to the reviewer’s comment.

  1. We have now revised Table II according to the reviewer’s comment and we have sorted the surveyed work according to the technique used.

  1. In the revised version of the manuscript, we have now largely expanded the application section (Section 6). This also include a larger description of hybrid and multiplexed technologies. Please, see lines 442 – 465.

  1. We have now revised the conclusion part according to the reviewer’s comment. Please, see lines 500-508.

Round 2

Reviewer 2 Report

This revised review paper seems much better for me to be published to the Micromachines.